# Exploring the Behavioral Intentions of Food Tourists Who Visit Crete

Georgios Angelakis [1,2,*], Yari Vecchio [3], Christos Lemonakis [1], Georgios Atsalakis [4], Constantin Zopounidis [4,5] and Konstadinos Mattas [6]

1   Department of Management Science & Technology, Hellenic Mediterranean University,
    72100 Agios Nikolaos, Greece; lemonakis@hmu.gr
2   Department of Business Economics & Management, CIHEAM-Mediterranean Agronomic Institute of Chania,
    73100 Chania, Greece
3   Department of Veterinary Medical Science, Alma Mater Studiorum University of Bologna,
    40126 Bologna, Italy; yari.vecchio@unibo.it
4   Financial Engineering Laboratory, Department of Production Engineering and Management,
    Technical University of Crete, 73100 Chania, Greece; gatsalakis@dpem.tuc.gr (G.A.);
    kostas@dpem.tuc.gr (C.Z.)
5   Institute of Finance, Audencia Business School, 44300 Nantes, France
6   Laboratory of Agricultural Products Marketing, Agricultural Policy and Cooperatives, Faculty of Agriculture,
    Aristotle University of Thessaloniki, 54124 Thessaloniki, Greece; mattas@auth.gr
*   Correspondence: angelakis@maich.gr

**Abstract:** Food tourism has been growing globally in recent years. Food tourism is considered as special interest tourism, attracting tourists who have a great interest in food. Tourists spend a significant percentage of their budget on the purchase of local food products and related food activities, contributing to the sustainable development of the touristic destination in the process. This survey took place in Crete, Greece, throughout the touristic period of 2021, and 4268 valid questionnaires were completed by international tourists. For the data analysis, the Structural Equation Model and an extended Theory of Planned Behavior Model, based on subjective norms, attitudes, perceived behavioral control, and satisfaction, were used to better understand the consumers' intentions to revisit and recommend the region of Crete. The outcomes of the research pinpointed that the perceived quality and perceived value of local foods positively influenced satisfaction, which, in turn, evoked favorable intentions to revisit and recommend Crete as a touristic destination. Moreover, while satisfaction, attitude, and subjective norms seem to be the most significant drivers affecting positive behavioral intentions, perceived behavior control seems to have had no significant impact. The implications and limitations of the survey, as well as future recommendations, are also discussed.

**Keywords:** food tourism; Theory of Planned Behavior; attitude; subjective norms; perceived behavioral control; satisfaction; motivational factors; behavioral intentions

## 1. Introduction

Gastronomy has become one of the fundamental components in the selection of a tourist destination. According to [1], among their main motivations in choosing their tourist destination, 15% of tourists are influenced by a place's gastronomy. A survey conducted by Hilton Worldwide found that roughly 36% of tourists visiting the Asia-Pacific region referred to food as a critical factor shaping the destinations to which they would travel [2]. According to the World Food Travel Association, 53% of leisure travelers are food travelers and 81% of travelers learn about food and drink when they visit a destination, while 59% of travelers believe that food and beverages are more important when they travel than they were 5 years ago [3]. This tendency seems to hold true in Greece during the pandemic as well, where tourists had a more positive attitude towards food than before COVID-19 and were more motivated to consume local food, resulting in spending more money on food

and being more willing to taste local food and visit Greek food establishments [4]. In this research, it has been proved that a significant percentage of tourists (31%) claimed that food experiences were very important in choosing Crete as their final destination.

Those trends have given rise to another type of tourism, so-called 'food tourism'. Tourists are identified according to different groups, such as comfort seekers, moderates, and authenticity seekers, based on the degree of interest in the authenticity of local food and similar gastronomic activities [5], and depending on the preference of their gastronomic experiences, they are identified as survivors, enjoyers, and experiencers [6]. The group of individuals whose primary reason for traveling is gastronomy, who are highly involved in related activities, are called gastronomes or food tourists/travelers, and they may even travel far away for the purpose of a food/gastronomic experience [7].

The contribution made by local products to local sustainable development has been recognized. By consuming local food products, tourists not only satisfy their vital needs but also interact with local culture and support local development by stimulating demand [8]. A more focused attitude on the local cuisine could have a major impact on higher planned expenditure on the part of tourists, with a maximum behavioral loyalty and greater appreciation of the quality or degree of innovation [9]. Ref. [10] concluded that local products were ranked in first place compared to global ones, with the most significance placed on their socioeconomic and health dimensions. Moreover, the Mediterranean Diet (MD) could be used as a dynamic tool for promoting sustainable development, emphasizing, among other elements, the mythological aspect of the MD for attaining cultural, economic, and social development [11].

Many studies have relied on food tourism characteristics to understand and measure the level of impact each or some of them have on various decisions such as destination and choice of food establishment or food-related behavior while on a vacation. According to [12], a conceptual Framework of Food Tourism Management was developed in order to understand the behavioral intentions of tourists, taking into account both the demand and supply perspective. The Theory of Planned Behavior (TPB), based on attitudes, subjective norms, and perceived behavior control, could be used for that purpose, but due to the multidimensional nature of the characteristics of food tourism from the demand and supply side at a destination, there is a need for an extended TPB, including more constructs in order to precisely identify and explain the behavioral intentions of tourists. In this paper, the behavioral intentions of food tourists are explored by using an extended Theory of Planned Behavior Model, taking into account the mediating role of satisfaction between perceived value and quality and behavioral intentions.

## 2. Literature Review

### 2.1. Motivational Factors

According to [13], there are two types of factors, namely push and pull, in any kind of travel decision. Push factors are internal motivations that urge tourists to decide to travel. Pull factors are external ones related to the destination's characteristics, which leads to making a decision about a holiday destination. Many studies are based on these motives when trying to understand and evaluate the level of influence they have on tourists' behavioral intentions while on vacation. Dimensions of motivation such as escape, change, excitement, taste of food/wine, socialization, interpersonal relationships, social status, family togetherness, relaxation, enjoyment, cultural experiences, novelty, knowledge, and learning can be characterized as push factors. Food tourism appeals, such as food events, food fairs, food trails and tours, markets, restaurants and cooking schools, food producers and staff, core wine products, wineries, food variety, and destination appeals (unique specialty shops, markets selling local farm produce, cultural events, the rural environment and farmers' markets, etc.) can be characterized as pull factors [14].

An exciting experience, food tourism appeal, interpersonal relationships, and sensory appeal significantly influence domestic tourists' intention to visit Bali, whereas the cultural experience, health concerns, and social values do not have significant effects on intentions

to revisit the region [15]. In the study of [16], it was found that high core wine attributes and education, high product involvement, and high participation in wine events, combined with escape and socialization or with the destination's attractiveness, lead to tourists' positive behavioral intentions in the Iberian Peninsula, while vineyard aesthetics, core wine attributes of the product, and educational experience are considered as the key drivers of wine tourism rather than the social context of the visit or the general regional characteristics in Northern Greece [17].

Based on the Experience Economy Model, the experiences connected with education and entertainment, and not the aesthetic and escapist dimensions, are the key factors for tourists who visit Cognac in France for wine tourism [18]. On the other hand, education and entertainment might not increase the tourists' perceptions of Street Food Festivals, while motivations related to escapism, memory, and aesthetics result in positive predictors [19]. According to the six-dimensional structure of experience quality, entertainment and learning factors were considered as the least important components [20]. Hedonic values influence consumers' intention to visit food trucks not only directly but also indirectly, through the creation of positive attitudes, while utilitarian values impact consumers' intention only indirectly by evoking a positive attitude toward food trucks [21]. Moreover, memorable local food experiences (cultural, educational, novelty, hedonism–meaningfulness, and adverse experience) can affect both tourists' intentions to recommend and to revisit a food establishment [20,22], while nostalgia has both direct and indirect effects on consumption intention in [23].

Concerning ethnic food in Malaysia, only when Dayak food is appreciated and accepted by the locals could the food be further promoted to tourists [24], whereas strong appreciation for local food specialties has a positive influence only on the intention to (re)visit the place of origin in mountainous areas in Italy [25]. Moreover, increasing the public's knowledge/awareness about festivals in the USA is regarded as a priority in order to have favorable behavioral intentions [26], while the interactive components of theme, product, and design were considered by tourists as key sources of learning and entertainment that enhance the attractiveness of a salt destination and influence tourists' decisions [27].

Although there are numerous studies focused on the above dimensions of push and pull factors as independent variables, the number of studies that investigate the possibility of perceived value and/or perceived quality as broader categories that directly or indirectly influence tourists' behavioral intentions is actually quite limited [28–33].

### 2.2. Visitor's Satisfaction

Gastronomic motivation affects the choice of the destination and the gastronomic experience influences satisfaction, while tourists express a high level of satisfaction with the gastronomic attributes of the destination, leading even to loyalty [1]. Experience quality has a significant effect on customers' loyalty, advocacy, and satisfaction; thus, pleasant experiences should be created in order to evoke higher satisfaction levels and to positively affect tourists' behavioral intentions [20]. Connecting satisfaction with the gastronomy of the destination may affect revisiting and/or recommending the place, given its culinary distinction [34]. This was the case with a festival in Malaysia, where the majority of tourists were satisfied and would not only revisit the festival again in the future but will recommend it to others as well [35]. Moreover, perceived value and perceived quality of a destination only indirectly affect the behavioral intention of tourists to recommend festivals in Taiwan through the positive influence on satisfaction [32]. In contrast, that outcome was not consistent with previous research [30,33]. Moreover, at halal-friendly destinations, the constructs of perceived value, destination satisfaction, trust, and loyalty significantly contribute to the prediction of tourists' future desire for a destination [36].

The elements of customers' emotional experience have to be comprehensively managed in order to achieve higher satisfaction that will enhance favorable behavioral intentions in halal food establishments [37]. The increase in friendly restaurant attachment

dimensions (experiential value) will increase experiential relationship quality dimensions (satisfaction and trust of customers), and this, in turn, will increase experiential relationship intentions [38]. The same results hold in the survey of [39] at pop-up restaurants, where those components led to an increase in behavioral outcomes, specifically the intention to spread positive word of mouth (WOM), return intentions, and an increased willingness to pay. Ref. [40] concluded that only one type of experiential value, namely, consumer return on investment (CROI), can positively and significantly affect a place's food image, while there were positive effects of prestige values (with the exclusion of uniqueness) on affective commitment, which, in turn, positively influences the customer's behavioral intention and their intention to revisit premium food markets [41].

In ethnic restaurants, authenticity has been considered a critical factor for enhancing customer satisfaction and purchase intentions, whereas the mediating role of customers' perceptions of authenticity is emphasized in which unfamiliar ingredients, unique food names, and stories about food origins enhance consumers' perceptions of authenticity [42]. Concerning perceived authenticity, unfamiliar food names, and ingredients significantly boost customers' perceptions of authenticity and evoke positive emotions, in comparison with familiar food names and ingredients that arouse negative emotions [43], whereas consumers' perceived authenticity affects their purchase intention not only directly but also indirectly through restaurant image and positive emotions [44].

Regarding wine tourism, two different attributes such as winery tours/wine tasting and overall ambiance impacted visitors' satisfaction, whereas the revisit intentions of winery tourists started to significantly decrease after a certain number of repeat visits, but it takes a higher number of repeat visits for highly satisfied tourists' revisit intentions to start diminishing [45]. There is also a positive relationship between wine promotion and customer satisfaction, while customer satisfaction significantly influences behavioral intention in South Korea [46]. Moreover, the effects of satisfaction with four types of local food (Portuguese, buffet, Michelin, and street-snack) and integrated satisfaction with food experience on WOM regarding Portuguese foods in Macau were investigated with different effects for satisfaction in each type [47]. Furthermore, the element of novelty-seeking produces moderate satisfaction with the consumption of local food. Refs. [48,49] suggested that visitors who prefer and take gastronomic experiences in Portugal seem to have neophilic tendencies (i.e., a preference for novel food flavours), given the fact that those groups of tourists (the so-called experiencers and authenticity seekers) have the tendency to seek out new food experiences [8].

Despite the fact that there is research focused on the above characteristics of the relations between emotional experiences, experiential values, perceived authenticity, promotion and satisfaction, there is a research gap in studies that investigate the possibility of perceived value and/or perceived quality as broader categories that indirectly influence tourists' behavioral intentions through satisfaction [28,31,32].

### 2.3. Theory of Planned Behavior

The Theory of Planned Behavior (TPB) is widely used in the food tourism sector. Three main characteristics are always evaluated in TPB, namely, attitudes, subjective norms and perceived behavior control, in order to examine their influence towards behavioral intentions to consume/purchase and WTP for local food products, to revisit the destination/food service establishment, and to recommend/use WOM about local food products, food establishments and destinations.

The study of [50] applies TPB to conclude that food's sensory appeal and the communication gap are not significant in the prediction of Chinese tourists' attitudes, whereas food concerns and table manners could deteriorate their attitudes in the consumption of unfamiliar local food in the USA. TPB is also used in the wine sector [51], suggesting that winescape service staff and complementary products in the first place, followed by winescape setting and wine value, had a significant influence on wine tourists' attitudes toward the winery, and, in turn, revisit intentions and willingness to recommend them. TPB

can be applied in the field of street food as well. Ref. [52] showed that affection, perceived service quality and satisfaction, rather than perception of hygiene, food quality, and value for money, have the highest influence on consumers' attitudes towards street food, which, in turn, affect their future behavioral intention in Phuket. Regarding ethnic food, using TPB, ref. [53] concluded that while tourists from different countries of Asia appear to have almost the same behavioral intentions, when they visit Sarawak in Malaysia, they have different consumption intentions toward Dayak food. TPB is also used in festivals. Attitude was the most significant component that influenced the revisit intentions of local and small culinary festivals in Italy with food and beverage quality in first place, and staff service and information had the greatest impact factor on tourists' attitudes; concerning the other two elements of TPB, subjective norms also affected behavioral intentions compared to perceived behavior control, which seems to have no statistical influence on revisit intentions [54]. These findings are in line with those of previous tourism studies [51,55,56].

An extended TPB model, including further major components in tourism marketing, was developed by [57]; besides the core TPB constructs, added variables such as travel motivation, eWOM, destination image, and destination familiarity indeed exert a significant impact on tourists' intentions to revisit Egypt, while a mediating role of attitude, subjective norms, and perceived behavioral control also exists. In their extended model of TPB incorporating satisfaction, destination image, perceived risk, service quality, and perceived value, ref. [28] showed that perceived behavioral control, perceived value, destination image, and satisfaction significantly influence visitors' revisit intentions, while perceived value, perceived service quality, and destination image had a strong impact on satisfaction, which, in turn, was found to be an important mediator between perceived service quality, destination image, and perceived value. In the extended model of [58], besides the fact that the three main constructs have a positive influence on domestic tourists' consumption intentions toward local food in China, it was also found that attitudes play a partial mediating role between benefit perceptions and intentions, with the benefit perceptions having a significant positive effect on attitudes and behavioral intentions, whereas risk perceptions affect behavioral intentions in a negative way.

By using perceived benefits to business, perceived difficulties in production, and the service of indigenous dishes, as well as customer patronage instead of perceived behavior control in the TPB extended model, ref. [59] found that attitude and perceived difficulties in production and the service of indigenous dishes were not significant predictors in affecting intentions to add more indigenous dishes, whereas subjective norms in the first place, followed by perceived customer patronage, were the most significant factors influencing behavioral intentions towards the inclusion of a greater variety of indigenous dishes on the menu of small- and medium-sized hotels in Ghana.

Ref. [60] developed an extended model of goal-directed behavior by including perceptions of authenticity, knowledge, and information search behavior in the original model of goal-directed behavior (MGB), which consisted of positive and negative anticipated emotions and frequency of past behavior, besides the elements of TPB that significantly influence desires, which, in turn, significantly influence tourists' behavioral intentions to visit a slow-tourism destination in Korea.

Although food tourists' behavioral intentions on some occasions could be explained by TPB, if there is a need to include broader characteristics of food tourism, an augmented TPB should be used. Characteristics that concern tourism marketing, perceived benefits, perceived risks and difficulties, positive and negative anticipated emotions, and perceptions of authenticity have already been included in extended TPB models [28,57–60]. However, there is a research gap when satisfaction is included as an extra component that directly affects the behavioral intentions of tourists, which also plays a mediator role between perceived quality, perceived value, and intentions [28].

## 3. Materials and Methods

The research took place at the International Airport of Chania in Crete, Greece, through a well-structured questionnaire and face-to-face interviews during the touristic period from May to September 2021. The questionnaire was developed based on a review of the literature on food tourism in order to better explore the drivers that affect behavioral intentions [8,32,54,58,60,61]. The questionnaire was translated into four languages: English, German, Italian, and Polish, and distributed to tourists through roll-ups, posters, and brochures who were waiting at the exit gates of the airport, having completed their vacations. Due to COVID-19 restrictions, the survey was conducted through an automated system of voluntary participation of foreign tourists using their personal electronic devices, in which the questionnaire was activated in the language of choice of the interested party through visual scanning of a suitable URL or QR Code from a PC, Smartphone, or Tablet. More than 5000 questionnaires were initially collected and a thorough screening and quality check of the data was made; some incomplete questionnaires were rejected, concluding with a sample size of 4268 total questionnaires that was finally considered as valid for analysis. In the year 2021, 662,000 tourist arrivals were recorded at the airport of Chania [62], so the sample size was quite large in order to extract statistically significant and trustworthy outcomes. Moreover, the sample size was representative, since nationalities were represented with percentages that were in line with the actual number of arrivals per nationality.

If we consider that since 2013, foreign tourist arrivals to Chania were half Scandinavian and the other half non-Scandinavian, in the last two years after the COVID period, this ratio has changed, with Scandinavians making up just one fourth of the total tourist arrivals, while Germans, the Polish, and the British occupy a percentage close to 40% of the total arrivals.

More specifically, in the year 2021, Germans took first place in total arrivals for the first time in recent years with 17%, displacing the British and the Swedish who were pushed to fourth and fifth place, with percentages of 11% and 7%, respectively. The significant increase in the Danish took them to second place with 16%, while the Polish maintained third place in terms of total arrivals with a percentage of 11%. The percentages of tourists from Norway and Finland is noteworthy: they recorded very low percentages, regressing towards the last positions in terms of total arrivals, when in previous years they were among the main nationalities visiting Western Crete.

In the literature, there is a dispute among researchers on sample size when the Structural Equation Model (SEM) is used [63]. Several academics suggest that outcomes derived from an analysis of no more than 200 questionnaires are not valid unless the community from which the sample was taken is also small [64]. Ref. [65] argues that a sample size of at least 50 observations is sufficient and that the rule of thumb that it should be more than 200 is too simplistic. Other researchers believe that 10 to 20 respondents per parameter estimate is sufficient [66]. The general rule is that the sample size depends always on the number of the real population size, the confidence level, and the margin of error to be set. In this research, the outcomes of the sample are considered trustworthy at a 99% confidence level with a 2% margin of error, or at a 95% confidence level with a 1.5% margin of error.

The items in this survey were measured through a five-point Likert-type scale, ranging from 1 = strongly disagree to 5 = strongly agree, in order to perform a SEM analysis. A total of 26 questions were answered by tourists in order to explore the drivers that affect behavioral intentions (Table 1).

SEM is a very general method that can be used to incorporate observable or latent variables (or factors) into structural models. The use of these models is widely used in the literature when one wants to identify facets of the behavior of individuals on a particular behavior or opinion, especially in food tourism studies [16,21,23,24,26,34,50,51,53,54,58,67].

**Table 1.** Questions regarding the constructs of the extended TPB.

| Questions/Variables | Dimension | Literature |
|---|---|---|
| Were you satisfied with the following characteristics of Cretan local products (freshness) ? (V1) | | |
| Were you satisfied with the following characteristics of Cretan local products (appearance) ? (V2) | | |
| Were you satisfied with the following characteristics of Cretan local products (taste) ? (V3) | Visitor's Satisfaction on Local Food Products | |
| Were you satisfied with the following characteristics of Cretan local products (quality) ? (V4) | | |
| Were you satisfied with the following characteristics of Cretan local products (price ? (V5) | | |
| Were you satisfied with the following characteristics of Cretan local products (varieties) ? (V6) | | |
| Eating local food products gives pleasure (V7) | | |
| Eating local products in its original place is exciting (V8) | Attitude | |
| Eating local products on holiday helps to relax (V9) | | |
| Eating local products evokes memories (V10) | | |
| It is important to me that the local products I eat on holiday look nice/taste good (V11) | | |
| I consume local products because they contain a lot of fresh ingredients produced in a local area (V12) | Perceived Quality | |
| Experiencing local products increases my knowledge about different cultures (V13) | | [8,32,54,58,60,61] |
| It is valuable for me to consume local products (V14) | | |
| Trying local food while traveling is very important for me (V15) | Perceived Value | |
| Tasting local products in an original place is an authentic experience for me (V16) | | |
| I like to talk to everybody about my local product experiences (V17) | | |
| Tasting local products enables me to have an enjoyable time with friends/family (V18) | Subjective Norms | |
| Most people who are important to me think I should consume local food when traveling (V19) | | |
| I am constantly sampling new and different food products (V20) | | |
| Whether or not I consume local food when traveling is completely up to me. (V21) | Perceived Behavioral Control | |
| I have time, money, and information to consume unfamiliar local food when traveling. (V22) | | |
| I intend to revisit Crete to savour its local products (V23) | | |
| I expect to consume more local food if I have a chance to revisit (V24) | | |
| I want to give advice about local product experiences to people who want to travel (V25) | Intention | |
| I am willing to recommend Cretan local products to others (V26) | | |

In Figure 1, the Structural Model is depicted together with the Hypotheses.

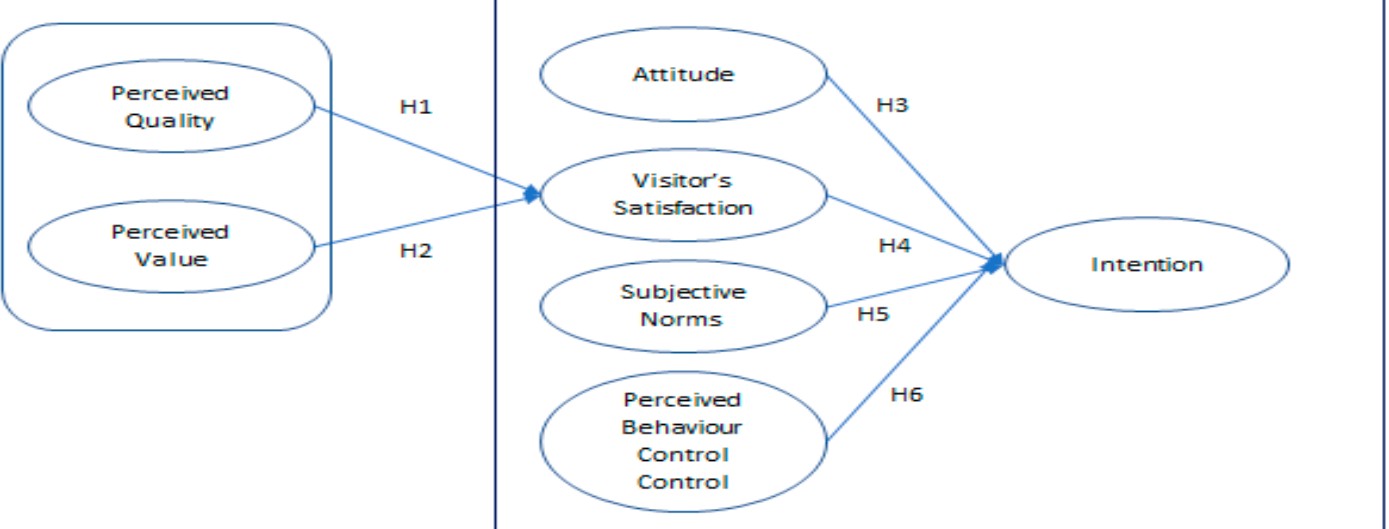

**Figure 1.** Extended TPB of this study.

The Hypotheses to be tested are the following:

**H1.** *Perceived quality is positively related to visitors' satisfaction.*

**H2.** *Perceived value is positively related to visitors' satisfaction.*

**H3.** *Attitude is positively related to behavioral intention.*

**H4.** *Visitors' satisfaction is positively related to behavioral intention.*

**H5.** *Subjective norms are positively related to behavioral intention.*

**H6.** *Perceived behavior control is positively related to behavioral intention.*

In order to use the SEM methodology to compare a given structure between groups, the invariant of the structure between groups must first be identified, followed by the use of an appropriate SEM technique. To evaluate the structure's invariant, all associated parameters must be set to the same value in a particular structure for all groups. When a limited SEM adequately matches the data, the supplied structure is invariant. Using the goodness of fit of an invariant structure, structural invariance between groups can be examined. In short, given a structure, the SEM approach could be used and considered to fit well according to the following: (1) the goodness of fit, depending on the *p*-value; and (2) the root mean square error of approximation (RMSEA), which considers the approximation error in a population and how well, if at all, the model would match the population covariance matrix. According to [68], RMSEA values $\leq 0.05$ can be considered as a good fit, values between 0.05 and 0.08 as an adequate fit, and values between 0.08 and 0.10 as a mediocre fit, whereas values >0.10 are not acceptable.

## 4. Results and Discussion

### 4.1. Descriptive Analysis Results

A descriptive analysis is depicted in Table 2, based on the distribution of the socio-demographic characteristics per different geographical area.

**Table 2.** Socio-demographic Characteristics.

| | | Eastern Europe | Central and North Europe | Scandinavian | Mediterranean | UK/ Ireland | USA/ Canada/ Australia | Other Countries | World |
|---|---|---|---|---|---|---|---|---|---|
| Age | 16–24 | 132 | 335 | 237 | 153 | 83 | 18 | 16 | 974 |
| | 25–34 | 351 | 498 | 232 | 207 | 153 | 41 | 48 | 1530 |
| | 35–44 | 183 | 205 | 101 | 94 | 82 | 21 | 32 | 718 |
| | 44–54 | 81 | 168 | 197 | 77 | 76 | 18 | 17 | 634 |
| | 55–64 | 8 | 78 | 142 | 35 | 47 | 10 | 8 | 328 |
| | 65 and over | 4 | 22 | 38 | 5 | 9 | 4 | 4 | 86 |
| Gender | Woman | 378 | 673 | 430 | 316 | 229 | 63 | 50 | 2139 |
| | Man | 381 | 630 | 512 | 251 | 217 | 48 | 75 | 2114 |
| Employment | Public employeee | 93 | 239 | 159 | 71 | 75 | 11 | 11 | 659 |
| | Private employee | 406 | 552 | 432 | 225 | 217 | 55 | 61 | 1948 |
| | Self employed | 129 | 135 | 72 | 102 | 60 | 22 | 31 | 551 |
| | Retired | 5 | 24 | 53 | 7 | 15 | 5 | 3 | 112 |
| | Student | 114 | 319 | 189 | 140 | 63 | 13 | 17 | 855 |
| Prefecture of Crete during your stay | Chania | 590 | 991 | 813 | 478 | 355 | 101 | 101 | 3429 |
| | Rethymno | 124 | 247 | 98 | 54 | 73 | 5 | 16 | 617 |
| | Heraklion | 30 | 42 | 16 | 21 | 17 | 3 | 5 | 134 |
| | Lasithi | 5 | 13 | 0 | 6 | 2 | 0 | 1 | 27 |
| Previous visit to Chania | No | 493 | 712 | 342 | 255 | 240 | 61 | 82 | 2185 |
| | Yes | 147 | 342 | 439 | 174 | 97 | 24 | 12 | 1235 |
| Type of acco-modation | Hotel | 414 | 625 | 777 | 233 | 231 | 51 | 64 | 2395 |
| | Rented apart-ment/room | 270 | 442 | 109 | 215 | 110 | 23 | 40 | 1209 |
| | Villas | 52 | 172 | 21 | 31 | 82 | 22 | 13 | 393 |
| | Friends | 13 | 31 | 7 | 49 | 12 | 7 | 1 | 120 |
| | Own house | 3 | 24 | 24 | 31 | 11 | 7 | 3 | 103 |
| | Camping | 2 | 4 | 0 | 5 | 0 | 1 | 2 | 14 |

Focusing on the distribution of specific variables, the sample is perfectly divided between men and women, in almost all geographical areas, while Crete is much more attractive to young people under 34 years old in all nationality groups as well as with the vast majority of the total population (59%) who prefer to spend holidays in the Greek island. Concerning employment, private employees represent 47% of the total population, whereas students represent quite a remarkable percentage with 21%, coming, particularly from Europe, which is also consistent with the data regarding age. The high number of young tourists is justified due to the fact that during the pandemic crisis, older people preferred not to travel in order not to be affected from COVID-19. The vast majority of tourists (82%) preferred to stay in the prefecture of Chania, rather than the other three prefectures of Crete, while 64% of foreign visitors visited Chania for the first time.

The fact that the majority of tourists chose to stay in Chania is because the research took place at the airport of Chania and not at the airport of Heraklion, which is a bigger city and serves a higher number of tourists who prefer to stay in that Prefecture of Crete; the fact that almost two out of three visitors visited the Prefecture of Chania for the first time is supported due to the fact that tourists from Scandinavian countries substantially decreased their arrivals due to restrictive measures from their countries, when they were traditionally

the repeaters; on the other hand, tourists from non-Scandinavian countries increased and there was also an increase in tourists from other new countries who were coming to Chania for the first time. Regarding the type of accommodation, hotels continued to be the first choice of tourists (57%), but there was an increasing demand for rented apartments/rooms (29%) and villas (9%). That increase is justified due to the fact that tourists preferred not to be in crowded spaces in a hotel and selected more independent places to stay with fewer people in order to reduce the possibility of being affected by COVID-19. Moreover, the fact that a significant percentage of tourists preferred to stay in villas (9%) indicates that they were of a high income level.

As far as local food products are concerned, tourists purchase different types of local food products during their stay with significant percentages (Table 3), such as olive oil (45.4%), wine (37.8%), cheese (34.9%), fresh orange juice (34.1%), raki/tsikoudia (local alcoholic drink) (33.6%), honey (29.7%), herbs and spices (28.7%), vegetables (26.4%), and rusk bread (16.9%).

**Table 3.** Purchase of local food products (%).

| Local Food Products | Frequency | Percent |
|---|---|---|
| Olive oil | 1963 | 45.4 |
| Wine | 1637 | 37.8 |
| Cheese | 1510 | 34.9 |
| Fresh orange juice | 1473 | 34.1 |
| Raki/Tsikoudia | 1453 | 33.6 |
| Honey | 1283 | 29.7 |
| Herbs and spices | 1241 | 28.7 |
| Vegetables | 1142 | 26.4 |
| Rusk bread | 730 | 16.9 |

*4.2. SEM Analysis Results*

The outcomes of the descriptive analysis of SEM are shown in Table 4. The high degree of tourists' satisfaction for local food attributes such as taste, quality, freshness, appearance, price, and varieties is evident.

**Table 4.** Descriptive statistics of SEM analysis results.

| Questions | Mean | Std. Deviation | Skewness | Kurtosis |
|---|---|---|---|---|
| | Statistic | Statistic | Statistic | Statistic |
| Were you satisfied with the following characteristics of Cretan local products (freshness)? | 4.49 | 0.602 | −0.831 | 0.327 |
| Were you satisfied with the following characteristics of Cretan local products (appearance)? | 4.41 | 0.631 | −0.765 | 0.529 |
| Were you satisfied with the following characteristics of Cretan local products (taste)? | 4.52 | 0.625 | −1.215 | 1.724 |
| Were you satisfied with the following characteristics of Cretan local products (quality)? | 4.46 | 0.628 | −0.949 | 1.087 |
| Were you satisfied with the following characteristics of Cretan local products (price)? | 4.17 | 0.780 | −0.854 | 0.862 |
| Were you satisfied with the following characteristics of Cretan local products (varieties)? | 4.15 | 0.790 | −0.834 | 0.804 |

**Table 4.** *Cont.*

| Questions | Mean | Std. Deviation | Skewness | Kurtosis |
| --- | --- | --- | --- | --- |
| | Statistic | Statistic | Statistic | Statistic |
| Eating local food products gives pleasure | 3.67 | 0.514 | −1.372 | 1.771 |
| Eating local food products in its original place is exciting | 3.55 | 0.591 | −1.028 | 0.629 |
| Eating local food products on holiday helps to relax | 3.29 | 0.701 | −0.644 | −0.143 |
| It is important to me that the local products I eat on holiday look nice/taste good | 3.64 | 0.531 | −1.142 | 0.696 |
| I consume local food products because they contain a lot of fresh ingredients produced in a local area | 3.51 | 0.544 | −0.485 | −0.777 |
| I like to talk to everybody about my local product experiences | 3.16 | 0.757 | −0.537 | −0.319 |
| Tasting local products enables me to have an enjoyable time with friends/family | 3.54 | 0.607 | −1.145 | 1.214 |
| It is valuable for me to consume local products | 3.62 | 0.546 | −1.146 | 0.758 |
| Trying local food while traveling is very important for me | 3.70 | 0.533 | −1.741 | 3.105 |
| I am constantly sampling new and different food products | 3.41 | 0.683 | −0.885 | 0.165 |
| Tasting local products evokes memories | 3.32 | 0.718 | −0.799 | 0.178 |
| Experiencing local products increases my knowledge about different cultures | 3.59 | 0.545 | −0.993 | 0.617 |
| Tasting local products in an original place is an authentic experience for me | 3.65 | 0.512 | −1.054 | 0.428 |
| Whether or not I consume local food when traveling is completely up to me. | 3.41 | 0.631 | −0.778 | 0.423 |
| I have the time, money, and information to consume unfamiliar local food when traveling. | 3.39 | 0.613 | −0.593 | 0.050 |
| Most people who are important to me think I should consume local food when traveling | 3.29 | 0.686 | −0.728 | 0.464 |
| I intend to revisit Crete to savour its local products | 3.63 | 1.209 | −0.602 | −0.519 |
| I expect to consume more local food if I have a chance to revisit | 3.48 | 0.632 | −0.946 | 0.417 |
| I want to give advice about local product experiences to people who want to travel | 3.19 | 0.753 | −0.639 | −0.019 |
| I am willing to recommend Cretan local products to others | 4.17 | 0.954 | −1.073 | 0.672 |

Thus, it is of utmost importance to include satisfaction in the extended Theory of Planned Behavior Model. Figure 2 shows the path diagram generated using the Stata 17. This path diagram can be employed to analyze the invariant of motivation structure, as given by Figure 2. The Chi square statistical value of 15,579.64 (with $p$ value 0.000) indicates the robust results of the model.

Revisiting the classical TPB model, adapted from Vesci and Botti's study [54], provided us with good results.

All the hypotheses underlying the model turned out to be significant. The perceived quality (H1) and value (H2) of local food both have a positive (0.22 and 0.31 coefficient, respectively) and significant influence ($p$-value less than 0.001) on visitors' satisfaction. This result confirms evidence from the literature on how perceptions of perceived value and perceived quality of product goodness influence personal satisfaction [8,31,32,54]; in turn, visitors' satisfaction (H4) was found to be the most important dimension (0.54 coefficient)

in influencing positively and significantly (*p*-value less than 0.001) the intention to revisit and recommend Crete. These findings are in line with previous research [32].

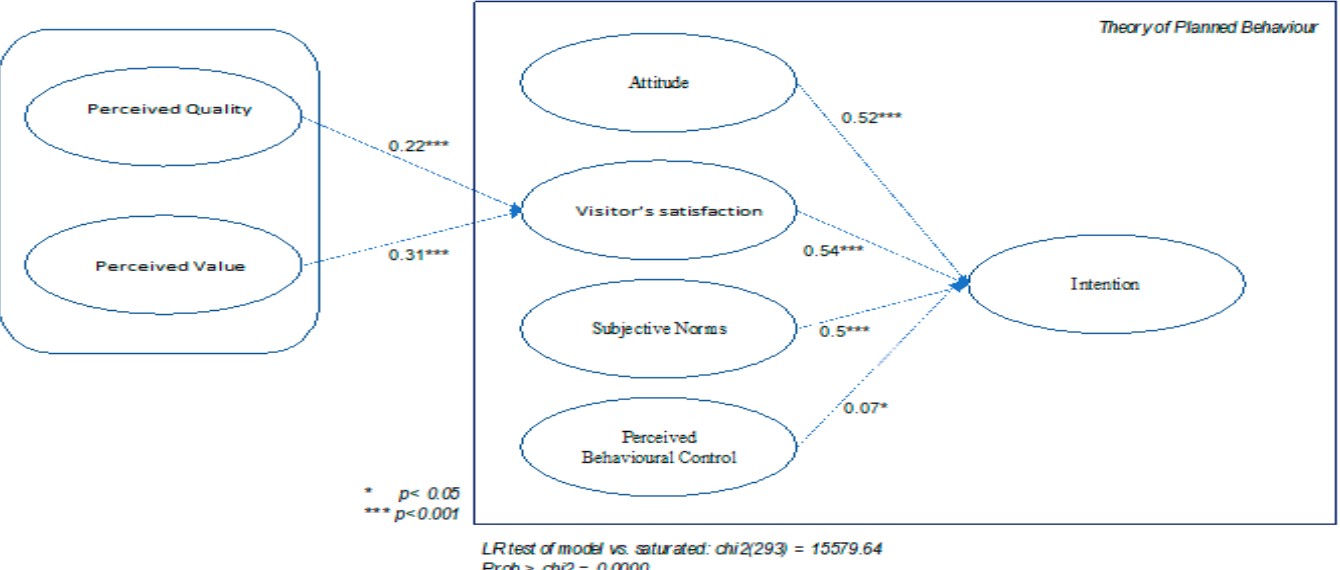

**Figure 2.** Results of the SEM analysis of the extended TPB.

Attitude (H3), Subjective Norms (H5), and Perceived Behavior Control (H6) are also related positively (0.52, 0.5, and 0.07 coefficient, respectively) and significantly to behavioral intentions. The three traditional elements of the TPB are significant in defining the intention to revisit Crete, this evidence is confirmed in the literature studies analyzed [32,58,60,61], showing how intentions to perform an action, in this case to return as a tourist, are influenced above all by variables located in the cognitive sphere of the person and not only by the context.

All tested relationships are significant with a *p*-value less than 0.001, apart from the construct of Perceived Behavioral Control, which seemed to be significant with a *p*-value less than 0.05 and a coefficient of 0.07 that has almost no effect on influencing behavioral intentions. These findings are almost in line with previous tourism studies [51,54–56].

### 4.3. Implications/Limitations of the Survey and Recommendations

The intentions of tourists to consume local food products, to revisit the destination location, and to recommend the place as a gastronomic destination could contribute to the sustainable development of the area [69–72]. The United Nations launched 17 Sustainable Development Goals (SDGs) that could be applied in all sectors of every nation [73]. Food tourism sustainability development refers mainly to economic, socio-cultural, and environmental dimensions, taking into account the involvement of tourists, local stakeholders, and local residents.

By consuming local products, there is less $CO_2$ produced in contrast to importing food from other countries; thus, environmental sustainability is supported. With the consumption and purchase of local food products, there are economic benefits that are derived for local producers, food establishments, and generally among the local society. This economic development results in an increase in employment in the agriculture sector as well as the touristic sector; social aspects are thus well advocated. With the consumption of local traditional food products, cultural aspects also emerge, since tourists could learn about the culture and traditions, the history of the destination, and its food—-knowledge and culture that can then be transferred, preserved, and protected.

Moreover, the involvement of local stakeholders is significant for the enhancement of sustainability. The creation of local networks of key stakeholders is of utmost importance for the effective promotion of the local food products and the touristic destination.

Coordination, voice mechanisms, knowledge sharing, mutual understanding [74], trust, and personal relationships [75] could contribute significantly to the internalization of small producers [76], the enhancement of innovation of local food products [77], and the provision of solutions for proximity difficulties in an urban target market where local industry is located in rural areas, which could be a competitive advantage compared to other touristic destinations.

The engagement of local people also plays a key role in the sustainable development of the area. Local people could act as part-time marketers through their interaction with tourists, evoking memorable food experiences [78], supporting participatory experiences [27], assisting in the transmission and preservation of knowledge attributes, culture, and production methods [79], taking part in food product tours [80], offering experiential activities to tourists [81], and creating unique tourist experiences with local food [82,83]. All of this could lead to the sustainable development of an area, which emphasizes the unique economic, cultural, and environmental characteristics of a place, increases the level of satisfaction, and creates favorable intentions to revisit and recommend the destination.

It is challenging to understand the motivational factors of various types of tourists, and the impact on behavioral intentions across a wide nexus of societies and culture. In this research, although there was a descriptive analysis per group of nationalities and sociodemograhic characteristics, the Structural Equation Model has been conducted based on the total number of tourists who visit Crete, savouring local food products and taking up related food activities. A segmentation of tourists according to nationality [84], age and gender [23,45,46], generation [39], cultural background [61,85], and education level [48] could be further implemented; as has been seen from previous researches, different outcomes on behavioral intentions are derived from different categories of tourists based on sociodemographic characteristics.

Moreover, this survey took place in 2021, one year after the outbreak of the pandemic, when it was difficult for older people to travel in order not to prevent themselves from being affected by COVID-19, and the majority of tourists were young (below 34 years old). Therefore, future research should also include more tourists aged 35 years and over, especially senior foodies and people with a high income level who are seeking to taste local food products [86]. Finally, concerning nationalities, a substantially decreased number of Scandinavian tourists were observed compared to periods before the pandemic due to COVID-19 restrictions that were imposed from their home countries.

Understanding ongoing changes in tourist drivers over time is crucial and will assist policy makers in the implementation of the appropriate marketing tools to enhance the satisfaction of tourists and stimulate their intentions to revisit and recommend the touristic destination. Successful marketing of a destination can be achieved by the establishment of an official website for visitors where they could upload their impressions [87,88]; by the creation of an official government Instagram account supporting the new trend of food-stagramming, which will advertise unique local food characteristics [89] that could also be very useful for tourists, since they could share food experiences, attitudes, and impressions from a destination that could influence the travel behavior of their followers [90]; by the promotion of annual events with national traditional foods [91]; and finally, by the promotion of material emphasizing the unique characteristics of Cretan gastronomy, combined with the culture and history of the area, and linking it with the healthy beneficial properties of the Mediterranean Diet.

## 5. Conclusions

According to the World Food Travel Association, food tourism is finally being considered as a mainstream travel experience since food has become a main motivation factor for travelers when choosing a destination, while spending more time and money on unique food and beverage experiences [92]. The so-called group of travelers known as food tourists, who are seeking to taste local food products and take part in related food activities while traveling, has been increasing substantially over recent years.

Previous research on food tourism has been conducted in an attempt to identify and include a range of motivational factors, but such studies have not yet fully explored the extent of the interwoven and multi-layered nature of tourist motives, based on push and pull factors from the demand and supply perspective of food tourism, given that no single model or theory of behavioral intentions can adequately explain all of them.

In this paper, the behavioral intentions of tourists were investigated with an extended Theory of Planned Behavior Model, including satisfaction as a mediator between perceived value, perceived quality, and behavioral intentions. It was found that perceptions of the value and quality of local food have a positive influence in the construct of visitors' satisfaction, which in turn positively affects the intentions to revisit and recommend Crete as a touristic destination. Moreover, satisfaction seemed to be the most important dimension together with attitude and subjective norms, whereas perceived behavior control, albeit having a positive and significant effect on behavioral intentions, does not have such a strong impact compared to the other components.

This paper aims to contribute to the existing food tourism research in Greece, specifically in the region of Crete, where tourism plays a vital and key role in the sustainable development of the island, since the contribution of tourism in the regional GDP is almost 50%. Food tourism is also a significant special interest tourism concern in Crete, since nearly one third of tourists who visit Western Crete considered food experiences as one of the main factors for choosing the island to spend their holidays.

The implications of COVID-19 were also evident, since the profile of tourists who visited Western Crete has changed compared to the pre-COVID period in terms of sociodemographic characteristics and travel behavior. Young tourists have increased, coming mainly from non-Scandinavian countries for the first time, with an increasing preference for touristic accommodation other than hotels, such as rented apartments and rooms, rented villas, and their own home or their friends' home, which could serve as a starting point for a regenerative, equitable, and inclusive form of tourism with an outlook [93].

The outcomes of this paper could be used as a guideline for relative stakeholders and decision makers in order to better promote the place as a food tourism destination. Initiatives should focus on enhancing the unique quality and the beneficial properties of local food attributes and food activities, connecting them with the culture and history of the place and enhancing knowledge by considering the emotional values of tourists and providing them with experiential authentic and memorable experiences. This will increase the degree of tourist satisfaction, stimulating positive intentions to revisit and recommend the destination.

Subsequent economic benefits could emerge that will consider Crete as an attractive and competitive food destination, which, together with successful marketing and management initiatives, could be implemented through the development of collective action [94] and a well-established cross-sector of Local Networks with effective coordination, mutual trust, and benefits among stakeholders. The active engagement of the local residents with tourism activities could contribute to the sustainable development of the destination by respecting the environment, encouraging local youth employment, tightening social relations, maintaining natural resources, and promoting cultural heritage.

This research should also consider some limitations of the study. A segmentation of tourists according to nationality, age and gender, generation, cultural background, and education level could be further implemented in future research.

**Author Contributions:** Conceptualization, G.A. (Georgios Angelakis), C.L., G.A. (Georgios Atsalakis) and C.Z.; methodology, G.A. (Georgios Angelakis), Y.V. and C.L.; software, G.A. (Georgios Angelakis) and Y.V.; validation, G.A. (Georgios Angelakis), Y.V., C.L., G.A. (Georgios Atsalakis), C.Z. and K.M.; formal analysis, G.A. (Georgios Angelakis), Y.V., C.L., G.A. (Georgios Atsalakis), C.Z. and K.M.; investigation, G.A. (Georgios Angelakis), Y.V. and C.L.; resources, G.A. (Georgios Angelakis); data curation, G.A. (Georgios Angelakis), Y.V. and C.L.; writing—original draft preparation, G.A. (Georgios Angelakis), Y.V. and C.L.; writing—review and editing, G.A. (Georgios Angelakis), Y.V., C.L., G.A. (Georgios Atsalakis), C.Z. and K.M.; visualization, G.A. (Georgios Angelakis) and C.L.;

supervision, C.L., G.A. (Georgios Atsalakis), C.Z. and K.M. All authors have read and agreed to the published version of the manuscript.

**Funding:** This research received no external funding.

**Institutional Review Board Statement:** The study was conducted in accordance with the Declaration of Helsinki and approved by the Research Ethical Committee of CIHEAM-Mediterranean Agronomic Institute of Chania (REC/2021/00858 on 12 May 2021).

**Informed Consent Statement:** Informed consent was obtained from all subjects involved in the study.

**Data Availability Statement:** The datasets created and/or analyzed during the current investigation are available upon reasonable request from the corresponding author.

**Acknowledgments:** We would like to thank all the respondents who patiently participated in the research.

**Conflicts of Interest:** The authors declare no conflict of interest.

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
