# Peer review of "Exploring the Behavioral Intentions of Food Tourists Who Visit Crete"

_sustainability, doi:10.3390/su15118961_

Round 1

Reviewer 1 Report

The study explored the behavioural intentions of food tourists who visit Crete. Conclusions from the conducted research are unambiguous and result from the obtained research results. The material used for the study is sufficient, and the research methods have been appropriately selected. The discussion of the results compared to other authors is very detailed. Publications cited by the authors of the article are well chosen. The authors mostly refer to the latest knowledge published in reputable scientific journals. I found no errors in the scientific aspect of the manuscript. Images quality can be replaced by a better quality one.

Acceptable. 

Reviewer 2 Report

1) Please deliberate on the need and justification for this research in Introduction Section
2) Please clarify the research gaps for each segment of literature review
3) Please explain about the sample design in detail in methodology section
4) Interpretations for each segment of analysis needs more clarity
5) Please also deliberate on the limitations of study
6) The conclusion can be improved

Please check the quality of English

Reviewer 3 Report

1- author need to rewrite the abstract and add the result

2- the analysis is not clear and completed

3- where is the mediation?

4- the data number is good but a bit old from 2021

5- need English improvement and

6- update some references

Need English improvement

Reviewer 4 Report

The manuscript  presents clearly and supported by scientific evidence concerning the Theory of Planned Behavior and the dimensions used in the model.

With regard to the methodology, it is important that the authors adjust the variables to the dimensions studied namely to the authors and studies referred to in the literature review, for example with regard to the pull and push motivation of tourists. For instance, the review mentions motivations and were not analysed, or should have made the connection to the dimensions studied, namely (Attitude - push motivations) (Perceived Quality; Perceived Value - both motivations) (Subjective Norms - push).

The SEM model was well chosen to run the data and answer the questions posed. A SEM that is typically used to test theory is applicable for this research because it can simultaneously specify the relationship between latent variables representing the theoretical concept in the proposed conceptual model of this research.

However, the statistical procedure for investigating relations between sets of observed and latent variables is that of factor analysis. There are two basic types of factor analyses: exploratory factor analysis (EFA) and confirmatory factor analysis (CFA), The authors should have  (CFA) that is appropriately used when the researcher has some knowledge of the underlying latent variable structure.

Moreover, considering the dimension of the sample a Cross Validation, in other words, the final best-fitting model for the calibration sample becomes the hypothesized model under test for the validation sample. In other words, if the adjusted model in the first sample shows a good fit in the second, it can be concluded that the model is invariant (maintains structure) on the two subsamples, and if these are representative of the population, the model is valid for the population which is the object of the study

With such a large sample it would make sense to add a cluster analysis by nationalities or by another category to measure behavioral intentions.

The authors could have further explored the data using SEM and further specified the measures of the model analysis.

In the Results and Discussion The response to the hypotheses are not supported by scientific evidences.

Implications and Recommendations presented are relevant and important for food tourism in Crete

Limitations and future research are not suggested.

In the Introduction chapter - 3rd paragraph where is” Moreover, the Mediterranean Diet (MD)…”put between parenthesis (MD) considering that you use this abbreviation afterwards.

2.3. Theory of Planned Behavior

2nd this subchapter in the 2nd paragraph substitutes “Regarding ethnic food, using the Theory of Planned Behavior…) by “Regarding ethnic food, using the TPB….” considering that it now uses the abbreviation TPB

Round 2

Reviewer 3 Report

ok

Reviewer 4 Report

The authors have made the requested changes.

Regarding the treatment of the data, the authors were careful to explain why they did not use other methods of analysis in this research.

Regarding limitations and future research, the authors added this aspect to the article.